# Distribution of Multidrug-Resistant Invasive Serotypes of *Streptococcus pneumoniae* during the Period 2007–2021 in Madrid, Spain

**DOI:** 10.3390/antibiotics12020342

**Published:** 2023-02-07

**Authors:** Sara de Miguel, Marta Pérez-Abeledo, Belén Ramos, Luis García, Araceli Arce, Rodrigo Martínez-Arce, Jose Yuste, Juan Carlos Sanz

**Affiliations:** 1Epidemiology Department, Directorate General of Public Health, Regional Ministry of Health of Madrid, 28002 Madrid, Spain; 2Department of Preventive Medicine, University Hospital 12 de Octubre, 28041 Madrid, Spain; 3CIBER of Respiratory Diseases (CIBERES), 28029 Madrid, Spain; 4Departamento de Epidemiología y Salud Pública, Epidemiología de las Enfermedades Infecciosas, Universidad de Alcalá, Alcalá de Henares, 28881 Madrid, Spain; 5Clinical Microbiology Unit, Public Health Regional Laboratory of the Community of Madrid, Directorate General of Public Health, Regional Ministry of Health of Madrid, 28055 Madrid, Spain; 6Spanish Pneumococcal Reference Laboratory, National Center for Microbiology, Instituto de Salud Carlos III, 28222 Madrid, Spain; 7CIBER of Epidemiology and Public Health (CIBERESP), 28029 Madrid, Spain

**Keywords:** *Streptococcus pneumoniae*, serotypes, multidrug-resistance, PCV13, penicillin, erythromycin, levofloxacin

## Abstract

After the systematic use of conjugate vaccines, the invasive pneumococcal disease (IPD) was included into the Madrid Notifiable Diseases Surveillance System through an Epidemiological Surveillance Network. Furthermore, *Streptococcus pneumoniae* was included in the Spanish Plan of Antibiotic Resistance. The aim of this study was to analyse the multidrug-resistant (MDR) phenotype distribution among invasive strains of *Streptococcus pneumoniae* isolated during 2007–2021 from usually sterile clinical samples in Madrid, Spain. A total number of 7133 invasive pneumococcal isolates were studied during the period from February 2007 to December 2021. Serotyping was characterised using the Pneumotest-Latex and by the Quellung reaction. Antibiotic susceptibility testing to penicillin (PEN), erythromycin (ERY), and levofloxacin (LVX) was performed using the E-test according to the EUCAST guidelines and breakpoints. Combination of non-susceptibility to PEN at standard dosing regimen (PNSSDR), resistance to ERY (ERYR) and to LVX (LVXR) was considered to be multidrug-resistant at standard dosing regimen of penicillin (MRPSDR), whereas the combination of resistance to PEN (PENR), ERYR, and LVXR was considered multidrug-resistant (MDR). The number of MDRPSDR and or MDR strains in the entire population (*n* = 7133) during the complete period (2007–2021) were 51 (0.7%) and 6 (0.1%), respectively. All MDRPSDR and/or MDR strains belonged to nine serotypes: 19A (*n* = 13), 15A (*n* = 12), 9V (*n* = 12), 14 (*n* = 7), 24F (*n* = 3), 15F (*n* = 1), 19F (*n* = 1), 6B (*n* = 1) and 6C (*n* = 1). Only two serotypes (9V and 19A) were found among MDR strains, and most of them (5/6) belonged to serotype 9V. Only 12.4% of the strains typified as serotype 9V were MDRPSDR and only 5.2% as MDR. The levels of pneumococcal MDRPSDR and/or MDR in this study were low and all six MDR strains were isolated between 2014 and 2018. These results reinforce the importance of monitoring the evolution of non-susceptible serotypes including those with MDR in the coming years, especially after the introduction of new conjugate vaccines of a broader spectrum.

## 1. Introduction

Antimicrobial resistance is becoming a global problem to public health systems worldwide [1]. The emergence of multidrug-resistant (MDR) *Streptococcus pneumoniae* has become a significant threat in recent years [2]. Because of this, this bacterium has been included in the Spanish Plan of Antibiotic Resistance (PRAN, according to its Spanish initials). The use of pneumococcal conjugate vaccines has affected the epidemiology and distribution of serotypes causing invasive pneumococcal diseases (IPD) [3,4,5] and has modified the trend in antibiotic resistance; this can influence the prevalence of MDR including the emergence of non-vaccine serotypes [6,7]. The 7-valent pneumococcal conjugate vaccine (PCV7) was introduced in the paediatric immunization program of the Community of Madrid (CM) in 2006. This vaccine included serotypes 4, 6B, 9V, 14, 18C, 19F and 23F. In 2010 it was replaced with the 13-valent pneumococcal conjugate vaccine (PCV13) that includes additionally the serotypes 1, 3, 5, 7F and 19A in 2010. However, this vaccine was removed from the paediatric immunization program in 2012 although it was partially used in the private market. Finally, PCV13 was introduced into the national immunization program for paediatric vaccination in 2015 with high coverage rates [8]. After the systematic use of conjugate vaccines, the invasive pneumococcal disease (IPD) was included into the Madrid Notifiable Diseases Surveillance System through an Epidemiological Surveillance Network. Doctors and microbiologists from public and private hospitals report the disease, providing clinical and microbiological information and strains for serotype characterization. Clinical and microbiological information, including the study of antibiotic resistance, is essential for the correct use of adequate antibiotic therapy. This would provide epidemiological surveillance to understand the behaviour of this disease and how it is adapting to the use of vaccines. Studying this behaviour allows for adopting measures to reduce the impact of this disease in the population, especially when affecting the most vulnerable patients.

The aim of this study was to analyse the MDR microbiological phenotype among invasive clinical strains of *S*. *pneumoniae* isolated from usually sterile clinical samples in the Madrid region during the period 2007 to 2021.

## 2. Results 

The number of MRPSDR and/or MDR in the strains studied (*n* = 7133) from the period 2007 to 2021 were 45 (0.6%) and 6 (0.1%), respectively. All MRPSDR and/or MDR strains belonged to nine serotypes: 19A (*n* = 13), 15A (*n* = 12), 9V (*n* = 12), 14 (*n* = 7), 24F (*n* = 3), 15F (*n* = 1), 19F (*n* = 1), 6B (*n* = 1) and 6C (*n* = 1) (Table 1). 

There was a significant correlation between certain serotypes and the development of reduced susceptibility to the antibiotics evaluated. Serotypes 19A, 15A, 9V, 14, 24F and 15F were associated with MRPSDR and/or MDR with a *p*-value < 0.05. Only two MDR serotypes (9V and 19A) were found and most of them (5/6) were serotype 9V, which showed a statistically significant association with displayed MDR (Table 2). Evaluation of MIC50 and MIC90 for PEN, ERY and LVX for serotype 9V strains, showed respectively values of a MIC50 = 2 mg/L and MIC90 = 3 mg/L for PEN, >256 mg/L and >256 mg/L for ERY and >32 mg/L and >32 mg/L for LVX. (Table 3).

Interestingly, each of the MDR strains was isolated between 2014 and 2018 with no detection during the early period after the introduction of PCV13. All MRPSDR cases of serotype 19A were isolated before 2016 and the only MDR case of our strain collection during the study period was isolated in the year 2014. Serotype 9V strains associated with MDR were isolated in 2014, 2015, 2017 and 2018, whereas MDRPSDR and/or MDR strains were distributed between 2010 and 2019. No cases of MDRPSDR and/or MDR were observed in the first two years of the pandemic period by SARS-CoV-2 (2020–2021) (Table 4).

## 3. Discussion

MDR pneumococci are usually defined as strains resistant to several classes of antimicrobials and diverse criteria have been applied to this definition. The results of the current study confirmed that levels of pneumococcal MRPSDR and/or MDR are low. In some studies, non-susceptibility (intermediate resistance or resistant) strains at standard dosing regimen to penicillin and other non-β-lactam antimicrobial classes including ≥2 antibiotic families [9,10] or even ≥3 families [11] have been considered. Among these antibacterial families, different antibiotics such as levofloxacin, erythromycin, clindamycin, tetracycline, trimethoprim–sulfamethoxazole and chloramphenicol have been included [9,11]. In the recommended antimicrobial therapies against *S*. *pneumoniae*, the preferred agents for non-PENR (MIC < 2 µg/mL) are β-lactam antibiotics and, alternatively, macrolides. Fluoroquinolones are not a first-line choice for penicillin-susceptible strains [2]. For PENR strains, the agents should be chosen based on their susceptibility, and include third generation cephalosporins and fluoroquinolones. Alternative antimicrobial agents to treat PENR strains include vancomycin or linezolid [2]. In many cases, the phenotype patterns of non-susceptibility to ≥3 antimicrobials included one of three combinations: erythromycin, tetracycline, and trimethoprim-sulfamethoxazole; penicillin, erythromycin, and trimethoprim-sulfamethoxazole; or penicillin, erythromycin, and tetracycline [10]. In fact, most MDR serotype studies focus on antibiotics that, besides β-lactams, include tetracycline, trimethoprim-sulfamethoxazole and chloramphenicol [12]. However, these agents are not of wide clinical use for IPD treatment. In contrast, fluoroquinolones and macrolides are more commonly employed as an alternative to β-lactams. For this reason, in the present study the selected agents for MDR categorization were erythromycin and levofloxacin. In this study, the profile of these three agents, which are among the most frequent antibiotics employed in clinical practice, was used for MRPSDR or MDR strains. Antimicrobial susceptibility testing for *S*. *pneumoniae* is usually recommended to be performed on Mueller–Hinton agar plus 5% defibrinated horse blood and 20 mg/L β-NAD (MH-F). However, Mueller–Hinton agar with 5% defibrinated sheep blood also provides an acceptable alternative for determining the MICs for *S*. *pneumoniae* [13] and therefore, this medium is valid when using the E-Test method [14]. This procedure has been employed in other antibiotic susceptibility studies [15]. Our study confirmed that 72.5% of the strains classified in these groups corresponded to serotypes 19A (25.5%), 15A (23.5%), and 9V (23.5%). However, although only a small proportion of serotype 19A strains were MRPSDR or MDR (2.3%), these percentages increased to 16.7%, 12.4% and 7.3% within serotypes 15F, 9V and 15A, respectively. Interestingly, it should be noted that these serotypes with a higher rate of antibiotic resistance have shown a case fatality rate between 10% and 20% in the same population [16]. The clinical isolates from this study have been characterized in previous studies published by our group describing the circulating serotypes confirming that in Madrid and Spain, serotype 8 was the most prevalent, followed by serotypes 3 and 22F, in all IPD cases in the adult population, although these serotypes are usually fully susceptible [16].

MDR to penicillin, tetracycline and macrolides of serotype 19A increased in Spain in the first decade of this century at the same time as an increase in serotype 19A incidence after the introduction of PCV7 that did not include this serotype [17,18]. After the use of PCV13, which already includes 19A, there was a decrease in the incidence of 19A and also an impact on antibiotic resistance with a reduction in the proportion of strains of this serotype with reduced susceptibility [18]. According to our results, serotype 19A, which has a demonstrated fatality of 12.5% [16], showed fewer resistant strains in the recent years of this study. This fact can be explained by the generic use of PCV13 in the paediatric population leading to a decrease in the prevalence of serotypes associated with antimicrobial resistance such as 19A [19]. 

In this study, most of the MDR strains were serotype 9V (five of the six MDR samples). This serotype has shown a lethality of 18,3%, which is similar to the lethality of serotype 3, one of the most prevalent serotypes [5,16]. These MDR isolates accounted for up to 5.2% of the strains of serotype 9V. Multidrug *S. pneumoniae* serotype 9V with reduced susceptibility or with resistance to penicillin and other agents such as macrolides and tetracycline has been described [20]. However, in our study it is important to remark that serotype 9V was also resistant to levofloxacin. This serotype has not been associated with levofloxacin resistance as frequently as it has been reported in other series [21]. The MIC90 to penicillin in this serotype was higher than the resistance breakpoint by EUCAST. However, this PCV13-covered serotype was not very frequently observed in our study (1.4% of all IPD) and the general prevalence of MDR 9V is very low [5]. One aspect of great interest is that we did not find MRPSDR and/or MDR in the past two years, probably due to the big decrease in IPD incidence caused by the COVID-19 pandemic. In this sense, a recent national study confirmed a generic rise in the proportion of resistant strains to different β-lactams and erythromycin, showing a marked increase in MIC90 to penicillin for serotype 11A [18]. However, in that study the authors did not evaluate multiple resistance to three agents as we did in our study (penicillin, erythromycin and levofloxacin). 

Serotype/serogroup-specific antibiotic resistance rates have been observed [22]. The mechanism underlying this phenomenon for β-lactam antibiotics in *S. pneumoniae* is based on mutations in the genes coding the penicillin-binding proteins (PBPs) [23]. The association of pili with widespread MDR genetic lineages of non-invasive paediatric *S. pneumoniae* isolates has also been documented [24]. Macrolide resistance is commonly due to ribosomal dimethylation by an enzyme encoded by erm(B), efflux by a two-component efflux pump encoded by mef(E)/mel msr(D) and, more rarely, mutations of the ribosomal target site of macrolides [25]. Fluoroquinolone resistance occurs by accumulated mutations within the bacterial genome, increased efflux, or acquisition of plasmid-encoded genes [26,27]. For instance, a recent article describes a virulent lineage of serotype 24F and identifies it as an emergent MDR pneumococcal variant [28]. In our series, three strains of this serotype were identified (2.17% of all IPD strains of this serotype). These strains were not fully genotyped, although further studies including genotyping analysis would be useful in the future. One aspect of interest is that MDR strains can be expanded worldwide by clonal propagation [29]. Hence, the main limitation of the present study is that we only performed the analysis at the phenotype level and therefore, we did not explore the molecular mechanisms of resistance and the genotypes involved.

An important benefit of using PCVs is their contribution to lowering the burden of antimicrobial resistance, by controlling serotypes that display reduced susceptibility [30] The increase in non-PCV13 serotypes associated with antibiotic resistance is concerning, especially the increase in penicillin resistance linked to serotypes 11A and 24F [18,31]. The future use of PCVs with an increasingly broad spectrum could reduce the impact of antibiotic resistance for non-PCV13 serotypes. In this sense, the use of PCV20 could prevent up to 93% of all pneumococcal isolates with reduced susceptibility to cefotaxime even during the pandemic period [18]. The results of our study reinforce the idea that it is essential to know circulating serotypes, their adaptation to new vaccines and the development of resistance, which represents a real challenge for the treatment and prevention of the most lethal serotypes.

## 4. Materials and Methods

*S*. *pneumoniae* strains isolated from IPD cases were sent to the Madrid Public Health Regional Laboratory from the microbiology services of public and private hospitals located throughout the entire Madrid region (Spain) in the period between February 2007 and December 2021. The strains were immediately tested after reception. For possible re-examinations, the isolates were conserved in skimmed milk at −80 °C. The samples analysed were mainly blood cultures, although any sterile sample that demonstrated the existence of IPD has been included in the study, such as cerebrospinal fluid, pleural fluid, and joint fluid. A total of 7133 invasive clinical isolates were studied for serotyping and monitored for antimicrobial susceptibility. Identification of the capsular serotypes was carried out using the Pneumotest-Latex (Statens Serum Institut, Copenhagen, Denmark) and the Quellung reaction using commercial antisera (Statens Serum Institut, Copenhagen, Denmark). Commercial strips of Benzylpenicillin, erythromycin and levofloxacin (ETEST^®^ strips; bioMérieux, Madrid, España S.A) with respective concentration ranks of 0.002–32 mg/L, 0.016–256 mg/L and 0.002–32 mg/L were used. The inoculum was adjusted to a bacterial concentration of 0.5 McFarland standard (or 1 McFarland standard for the mucoid strain) and the *S*. *pneumoniae* ATCC 49619 was employed as a reference strain. The strips were applied to the surface of the inoculated agar plates (Mueller Hinton 2 agar + 5% sheep blood [MHS], bioMérieux España, S.A.) and incubated at 35 ± 2 °C in a 5% CO_2_ atmosphere during 20 to 24 h. MIC values were read from the scale at the intersection point between the complete inhibition ellipse edge and the strip. Interpretation of susceptibility results was established according to the EUCAST breakpoints [32,33]. Strains showing a minimum inhibitory concentration (MIC) to penicillin >2 mg/L were considered penicillin-resistant (PENR) and with MIC > 0.06 mg/L were categorized as non-susceptible at standard dosing regimen (PNSSDR). Strains with MIC > 0.5 mg/L to erythromycin and MIC > 2 mg/L to levofloxacin were considered resistant to erythromycin (ERYR) and levofloxacin (LVXR), respectively. Pneumococcal strains assigned as PNSSDR, ERYR and LVXR were evaluated as multidrug-resistant at standard dosing regimen of penicillin (MRPSDR) and the combination of PENR, ERYR, and LVXR, was considered as MDR. The MIC values corresponding to inhibition of ≥50% or ≥90% of the strains among the total number of isolates were obtained and expressed as MIC50 and MIC90. To calculate the statistical association between serotypes and MRPSDR and/or MDR during the complete period (2007–2021, the Odds Ratios (OR) with its correspondent 95% confidence intervals (CI95) were calculated. The statistical significance was set at *p*-value < 0.05. Statistical analyses were performed using STATA v.16.

## Figures and Tables

**Table 1 antibiotics-12-00342-t001:** Pneumococcal serotypes with multidrug-resistant at standard dosing regimen of penicillin and/or multidrug-resistant in the Madrid Region during the period 2007 to 2021.

Serotype	IPD (*n*)	MRPSDR and/or MDR (*n*)	% Respecting All Strains MRPSDR and/or MDR	% Respecting All IPD Strains of This Serotype	OR (CI95)
19A	567	13	25.49	2.29	4 (2.1–7.6)
15A	164	12	23.53	7.32	14 (7.2–27.3)
9V	97	12	23.53	12.37	25.3 (12.8–50.1)
14	156	7	13.73	4.49	7.4 (3.3–16.7)
24F	138	3	5.88	2.17	3.2 (1–10.5)
15F	6	1	1.96	16.67	28.3 (3.2–246.7)
19F	107	1	1.96	0.93	1.3 (0.2–9.6)
6B	32	1	1.96	3.13	4.5 (0.6–34)
6C	194	1	1.96	0.52	0.7 (0.1–5.2)

IPD: Invasive pneumococcal diseases; MRPSDR: Multidrug-resistant at standard dosing regimen; MDR: Multidrug-resistant; OR: Odds Ratio.

**Table 2 antibiotics-12-00342-t002:** Multidrug-resistant pneumococcal serotypes in the Madrid Region during the period 2007 to 2021.

Serotype	IPD (*n*)	MDR (*n*)	% Respecting All Strains MDR	% Respecting All IPD Strains of This Serotype	OR (CI90)
9V	97	5	83.3	5.2	387 (44.7–3342)
19A	567	1	16.7	0.2	2.35 (0.27–20.1)

IPD: Invasive pneumococcal diseases; MDR: Multidrug-resistant; OR: Odds Ratio.

**Table 3 antibiotics-12-00342-t003:** MIC50 and MIC90 of PEN, ERY and LVX for all IPD cases and for multidrug-resistant strains at standard dosing regimen of penicillin and/or MDR serotypes in the Madrid Region for the period 2007 to 2021.

Strains/Serotypes	*n*	% Respecting All IPD	PEN MIC Rank	PEN MIC_50_ (mg/L)	PEN MIC_90_ (mg/L)	ERY MIC Rank (mg/L)	ERY MIC_50_ (mg/L)	ERY MIC_90_ (mg/L)	LVX MIC rank (mg/L)	LVX MIC_50_ (mg/L)	LVX MIC_90_ (mg/L)
All IPD 2007–2021	7133	100.00	<0.002–12	0.02	0.75	<0.016–>256	0.125	>256	0.19–>32	1	2
All IPD 2007–2021 MRPSDR and/or MDR	51	0.71	0.094–6	1	3	>256	>256	>256	3–>32	>32	>32
19A MRPSDR and/or MDR	13	0.18	0.094–4	0.75	2	>256	>256	>256	3–>32	>32	>32
15A MRPSDR and/or MDR	12	0.17	0.19–1.5	1	1.5	>256	>256	>256	3–>32	>32	>32
9V MRPSDR and/or MDR	12	0.17	0.5–6	2	3	>256	>256	>256	>32	>32	>32
14 MRPSDR and/or MDR	7	0.10	0.5–2	2	2	>256	>256	>256	3–>32	>32	>32
24F MRPSDR and/or MDR	3	0.04	0.5–1	0.75	1	>256	>256	>256	3	3	3
15F MRPSDR and/or MDR	1	0.01	0.19	0.19	0.19	>256	>256	>256	3	3	3
19F MRPSDR and/or MDR	1	0.01	0.25	0.25	0.25	>256	>256	>256	>32	>32	>32
6B MRPSDR and/or MDR	1	0.01	1.5	1.5	1.5	>256	>256	>256	>32	>32	>32
6C MRPSDR and/or MDR	1	0.01	1.5	1.5	1.5	>256	>256	>256	4	4	4

IPD: Invasive pneumococcal diseases (IPD); MIC: Minimum inhibitory concentration; MIC_50_: Minimum inhibitory concentration values corresponding to inhibition of ≥50% of the strains; MIC_90_: Minimum inhibitory concentration values corresponding to inhibition of ≥50% of the strains; PEN: Penicillin; ERY: Erythromycin; LVX: Levofloxacin; MRPSDR: Multidrug-resistant at standard dosing regimen; MDR: Multidrug-resistant.

**Table 4 antibiotics-12-00342-t004:** Pneumococcal serotypes with multidrug-resistant at standard dosing regimen of penicillin and/or multidrug-resistant serotypes by year in the Madrid Region for the period 2007 to 2021.

Year	Invasive Strains (*n*)	MDR	MRPSDR	19A (*n*)	15A (*n*)	9V (*n*)	14 (*n*)	24F (*n*)	15F (*n*)	19F (*n*)	6B (*n*)	6C (*n*)
2007–2021	7133	6	45	13	12	12	7	3	1	1	1	1
2007	539	0	1	1	0	0	0	0	0	0	0	0
2008	710	0	4	2	0	0	0	0	0	1	1	0
2009	730	0	3	1	0	0	2	0	0	0	0	0
2010	482	0	4	1	1	1	0	0	1	0	0	0
2011	466	0	2	2	0	0	0	0	0	0	0	0
2012	366	0	3	2	1	0	0	0	0	0	0	0
2013	331	0	7	1	0	2	2	1	0	0	0	1
2014	394	2	5	1	2	1	1	2	0	0	0	0
2015	468	1	8	2	4	2	1	0	0	0	0	0
2016	504	0	1	0	1	0	0	0	0	0	0	0
2017	548	2	1	0	0	3	0	0	0	0	0	0
2018	591	1	2	0	1	1	1	0	0	0	0	0
2019	633	0	4	0	2	2	0	0	0	0	0	0
2020	210	0	0	0	0	0	0	0	0	0	0	0
2021	161	0	0	0	0	0	0	0	0	0	0	0
IPD (*n*)				567	164	97	156	138	6	107	32	194

## Data Availability

Not applicable.

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
