# Peer review of "Distribution of Multidrug-Resistant Invasive Serotypes of *Streptococcus pneumoniae* during the Period 2007–2021 in Madrid, Spain"

_antibiotics, 2023, doi:10.3390/antibiotics12020342_

Round 1
Reviewer 1 Report
A good capacious and brief scientific article Serotypes of multidrug-resistant pneumococcus in the Madrid region in the period from 2007 to 2021. There is nothing unnecessary. An excellent article summarizing the research of many years
Author Response
We appreciate and we receive all suggestions with interest to improve our work. We hope that this work will be interesting for publication in this journal.
Best regards
Reviewer 2 Report
After reviewing the manuscript entitled: Characterization of multidrug-resistant invasive serotypes of Streptococcus pneumoniae during the period 21007-2021 in Madrid, Spain, the following comments should be considered:
1- The title should be modified
2- A short background should be added to the abstract.
3- What is the basis of the selection of the utilized antibiotics?
4- the introduction is too short. More details should be added to it to be more comprehensive.
5- Why the genotypic aspect wasn't studied in this manuscript?/
6- Table 1 is not clear and it is better to represent it in a bar chart.
7- The detailed method of MIC determination should be mentioned.
8- what is the method of preserving the isolates?
9- Table 4 is not clear
Author Response
We have attached below the comments and also as a word attachment in case it is easier for you to review:
Dear Reviewers,
It is a pleasure for us to have the opportunity to present our results in this journal. The last few years have been hard due to the Covid pandemic but despite everything, we have managed to continue working on invasive pneumococcal disease (IPD). This important disease causes a high number of people affected all over the world and produces serious lifethreatening conditions due in part to the worrying increase in resistance to penicillin and other antibiotics that has occurred in the last few decades. It is crucial, especially in these times in which pneumonia has become very important in our daily lives, to prevent IPD by using vaccines that we have on the market and newer vaccines of broader spectrum are coming. During the SARS-CoV-2 pandemic, COVID patients were treated with antibiotics as a preventive measure for potential co-infections and secondary infections. The effect of this treatment on the antimicrobial resistance rates of S. pneumoniae and even other respiratory pathogens is not fully understood. We consider that our results are interesting since they show the evolution of antibiotic resistance in recent years and allow us to clarify in serotypes, we should focus our preventive efforts on, adapt the vaccination schedule to the epidemiological situation, and focus on vaccine development.
We appreciate and we receive all suggestions with interest to improve our work. We hope that all the answers and changes made are to your liking and we hope that this work will be interesting for publication in this journal.
Best regards
REVIEW
After reviewing the manuscript entitled: Characterization of multidrug-resistant invasive serotypes of Streptococcus pneumoniae during the period 2007-2021 in Madrid, Spain, the following comments should be considered:
1.The title should be modified
The title has been slightly modified
Distribution of multidrug-resistant invasive serotypes of Streptococcus pneumoniae during the period 21007-2021 in Madrid, Spain
2- A short background should be added to the abstract.
A brief background has been be added at the beginning of the abstract:
After the systematic use of conjugate vaccines, invasive pneumococcal disease (IPD) was included into the Madrid Notifiable Diseases Surveillance System through an Epidemiological Surveillance Network. Further, Streptococcus pneumoniae was included in the Spanish Plan of Antibiotic Resistance.
3- What is the basis of the selection of the utilized antibiotics?
Commentaries and a new references have been introduced in the discussion section:
Most of MDR serotype studies are focused to antibiotics that, besides β-lactams, includes other families as tetracycline, trimethoprim-sulfamethoxazole and chloramphenicol (Patil 2022).
Patil S, Chen H, Lopes BS, Liu S, Wen F. Multidrug-resistant Streptococcus pneumoniae in young children. Lancet Microbe. 2022 Oct 31:S2666-5247(22)00323-8. doi: 10.1016/S2666-5247(22)00323-8. Epub ahead of print. PMID: 36328027.
However, these agents are not of wide clinical use for IPD treatment. In contrast, fluoroquinolones and macrolides are more commonly employed as an alternative to β-lactams. For this reason, in the present study the selected agents for MDR categorization were erythromycin and levofloxacin.
A new sentence related to the serotype 9V and levofloxacin resistance and three new cites has been added to the discussion:
Multidrug S. pneumoniae serotype 9V with reduced susceptibility or with resistance to penicillin and other agents as macrolides and tetracycline has been described (Jauneikaite 2017). However, in our study is important to remark that serotype 9V was also resistant to levofloxacin. This serotype has not been associated to levofloxacin resistance as frequent as it has been reported in other series (Baek 2018).
Jauneikaite E, Khan-Orakzai Z, Kapatai G, Bloch S, Singleton J, Atkin S, Shah V, Hatcher J, Samarasinghe D, Sheppard C, Fry NK, Satta G, Sriskandan S. Nosocomial Outbreak of Drug-Resistant Streptococcus pneumoniae Serotype 9V in an Adult Respiratory Medicine Ward. J Clin Microbiol. 2017 Mar;55(3):776-782. doi: 10.1128/JCM.02405-16. Epub 2016 Dec 14. PMID: 27974539; PMCID: PMC5328445.
Baek JY, Kang CI, Kim SH, Ko KS, Chung DR, Peck KR, Lee NY, Song JH. Emergence of multidrug-resistant clones in levofloxacin-nonsusceptible Streptococcus pneumoniae isolates in Korea. Diagn Microbiol Infect Dis. 2018 Jul;91(3):287-290. doi: 10.1016/j.diagmicrobio.2018.02.010. Epub 2018 Feb 16. PMID: 29540263.
4- the introduction is too short. More details should be added to it to be more comprehensive.
The next sentence has been added to the introduction
After the systematic use of conjugate vaccines, the invasive pneumococcal disease (IPD) was included into the Madrid Notifiable Diseases Surveillance System through an Epidemiological Surveillance Network. Doctors and microbiologists from public and private hospitals, report the disease, providing clinical and microbiological information and strains for serotype characterization.
5- Why the genotypic aspect wasn't studied in this manuscript?
In the discussion section we indicate: “Hence, the main limitation of the present study is that we only performed the analysis at the phenotype level and therefore, we did not explore the molecular mechanisms of resistance and the genotypes involved”.
To emphasize this limitation new commentaries and references have been added:
The association of pili with widespread MDR genetic lineages of non-invasive pediatric Streptococcus pneumoniae isolates has also been documented (Alexandrova 2022).
Alexandrova AS, Pencheva DR, Setchanova LP, Gergova RT. Association of pili with widespread multidrug-resistant genetic lineages of non-invasive pediatric Streptococcus pneumoniae isolates. Acta Microbiol Immunol Hung. 2022 Sep 12. doi: 10.1556/030.2022.01816. Epub ahead of print. PMID: 36094859.
For instance, a recent article describing a virulent lineage of serotype 24F has been identified as an emergent MDR pneumococcal variant (Lo 2022). In our series, three strains of this serotype were identified (2.17% of respecting all IPD strains of this serotype). These strains were not fully genotyped although further studies including genotyping analysis would be useful in the future.
Lo SW, Mellor K, Cohen R, Alonso AR, Belman S, Kumar N, Hawkins PA, Gladstone RA, von Gottberg A, Veeraraghavan B, Ravikumar KL, Kandasamy R, Pollard SAJ, Saha SK, Bigogo G, Antonio M, Kwambana-Adams B, Mirza S, Shakoor S, Nisar I, Cornick JE, Lehmann D, Ford RL, Sigauque B, Turner P, Moïsi J, Obaro SK, Dagan R, Diawara I, Skoczyńska A, Wang H, Carter PE, Klugman KP, Rodgers G, Breiman RF, McGee L, Bentley SD, Muñoz-Almagro C, Varon E; Global Pneumococcal Sequencing Consortium. Emergence of a multidrug-resistant and virulent Streptococcus pneumoniae lineage mediates serotype replacement after PCV13: an international whole-genome sequencing study. Lancet Microbe. 2022 Oct;3(10):e735-e743. doi: 10.1016/S2666-5247(22)00158-6. Epub 2022 Aug 16. PMID: 35985351; PMCID: PMC951946
6- Table 1 is not clear and it is better to represent it in a bar chart.
We thank the Reviewer for the suggestion. The table contains many data and information that it is not possible to represent it using a figure with bars. The table contains different serotypes, the number of IPD cases per serotype, the number of strains with reduced susceptibility or resistance, the proportion of strains and the Odds-ratio. Designing a figure containing all these parameters associated to several serotypes would be even more difficult to interpret.
7- The detailed method of MIC determination should be mentioned.
To respond these questions, the following sentence has been included in materials and methods:
Commercial strips of Benzylpenicillin, erythromycin and levofloxacin (ETEST® strips; bioMérieux España, S.A.) with respective concentrations ranks of 0.002-32 mg/L, 0.016-256 mg/L and 0.002-32 mg/L were used. The inoculum was adjusted to a bacterial concentration of 0.5 McFarland standard (or 1 McFarland standard if mucoid strain) and the S. pneumoniae ATCC 49619 was employed as reference strain. The strips were applied to the surface of the inoculated agar plates (Mueller Hinton 2 agar + 5% sheep blood [MHS], bioMérieux España, S.A.) and incubated at 35±2 °C in a 5%CO2 atmosphere during 20 to 24 h. MIC values were read from the scale at the intersection point between the complete inhibition ellipse edge and the strip.
8- what is the method of preserving the isolates?
We have included a sentence in Material and Methods to explain:
The strains were immediately tested after reception. For possible reexaminations, the isolates were conserved in skimmed milk at -80ºC.
9- Table 4 is not clear
We thank the reviewer for his suggestion. We have made changes to the table to make it clearer

Reviewer 3 Report
Although the manuscript is of interest and data are well presented (even with the disclosed limitations), these analyses suffer from an important mistake. According to EUCAST website and Etest application guide, antimicrobial susceptibility testing for Streptococcus pneumoniae should be stated on Mueller-Hinton agar + 5% defibrinated horse blood and 20 mg/L β-NAD (MH-F).
The introduction should specify which Streptococcus pneumoniae serotypes are covered by which vaccine types. This is important for the proper analysis of the course of microbiological phenomena.
Minor revision:
The method of writing percentage values should be unified, i.e. 12.5% instead 12,5%, and ‘one place after the dot’ in Table 1.
Editorial errors in lines: 36, 49, 54, 84, 96, 133, 171, 175.
The abbreviations used in the tables should be explained below.
Author Response
We have attached below the comments and also as a word attachment in case it is easier for you to review:
Dear Reviewers,
It is a pleasure for us to have the opportunity to present our results in this journal. The last few years have been hard due to the Covid pandemic but despite everything, we have managed to continue working on invasive pneumococcal disease (IPD). This important disease causes a high number of people affected all over the world and produces serious lifethreatening conditions due in part to the worrying increase in resistance to penicillin and other antibiotics that has occurred in the last few decades. It is crucial, especially in these times in which pneumonia has become very important in our daily lives, to prevent IPD by using vaccines that we have on the market and newer vaccines of broader spectrum are coming. During the SARS-CoV-2 pandemic, COVID patients were treated with antibiotics as a preventive measure for potential co-infections and secondary infections. The effect of this treatment on the antimicrobial resistance rates of S. pneumoniae and even other respiratory pathogens is not fully understood. We consider that our results are interesting since they show the evolution of antibiotic resistance in recent years and allow us to clarify in serotypes, we should focus our preventive efforts on, adapt the vaccination schedule to the epidemiological situation, and focus on vaccine development.
We appreciate and we receive all suggestions with interest to improve our work. We hope that all the answers and changes made are to your liking and we hope that this work will be interesting for publication in this journal.
Best regards
REVIEW:
Although the manuscript is of interest and data are well presented (even with the disclosed limitations), these analyses suffer from an important mistake. According to EUCAST website and Etest application guide, antimicrobial susceptibility testing for Streptococcus pneumoniae should be stated on Mueller-Hinton agar + 5% defibrinated horse blood and 20 mg/L β-NAD (MH-F).
The purpose of the medium employed in this study (Mueller Hinton agar + 5% sheep blood [MHS], bioMérieux España, S.A.) is to detect the antimicrobial susceptibility of pneumococci and other streptococci to antibiotics. According to the manufacturer, this medium enables the growth of theses bacteria, and guarantee the minimum interference from the formula constituents in the results of the antimicrobial susceptibility test. However, to answer the Reviewer, a sentence has been introduced in the discussion section:
Antimicrobial susceptibility testing for S. pneumoniae is usually recommended to be performed on Mueller-Hinton agar plus 5% defibrinated horse blood and 20 mg/L β-NAD (MH-F). However, Mueller-Hinton agar with 5% of defibrinated sheep blood also provides an acceptable alternative for determining the MICs for S. pneumoniae (D'Amato 1987) and therefore, this medium is valid when using the E-Test method (Jorgensen 1994). This procedure has been employed in other antibiotic susceptibility studies (Pantosi 2009)
Three new references have been added
D'Amato RF, Swenson JM, McKinley GA, Hochstein L, Wallman AA, Cleri DJ, Mastellone AJ, Fredericks L, Gonzalez L, Pincus DH, et al. Quantitative antimicrobial susceptibility test for Streptococcus pneumoniae using inoculum supplemented with whole defibrinated sheep blood. J Clin Microbiol. 1987 Sep;25(9):1753-6. doi: 10.1128/jcm.25.9.1753-1756.1987. PMID: 3654946; PMCID: PMC269321.
Jorgensen JH, Ferraro MJ, McElmeel ML, Spargo J, Swenson JM, Tenover FC. Detection of penicillin and extended-spectrum cephalosporin resistance among Streptococcus pneumoniae clinical isolates by use of the E test. J Clin Microbiol. 1994 Jan;32(1):159-63. doi: 10.1128/jcm.32.1.159-163.1994. PMID: 8126173; PMCID: PMC262988.
Pantosti A, D'Ambrosio F, Tarasi A, Recchia S, Orefici G, Mastrantonio P. Antibiotic susceptibility and serotype distribution of Streptococcus pneumoniae causing meningitis in Italy, 1997-1999. Clin Infect Dis. 2000 Dec;31(6):1373-9. doi: 10.1086/317502. Epub 2000 Nov 29. PMID: 11096005.
The introduction should specify which Streptococcus pneumoniae serotypes are covered by which vaccine types. This is important for the proper analysis of the course of microbiological phenomena.
We agree with the reviewer's suggestion. We have added some sentences in the introduction that explain the serotypes included in each vaccine.
Minor revision:
We thank the reviewer for his suggestion. We have made changes to unify the method of writing percentage values
Editorial errors in lines: 36, 49, 54, 84, 96, 133, 171, 175.
Changes have been made to fix the editorial errors. We thank the reviewer.
The abbreviations used in the tables should be explained below.
We thank the Reviewer for his suggestion. The explanation of the abbreviations used have been added at the bottom of the tables.

Reviewer 4 Report
I consider that the paper can be published in the current format with minimal changes and spelling corrections. It would be advisable to describe the typology of the samples from which the bacterial clinical isolates were obtained. I also recommend including minimal information on the serotype of the 5672 isolates not included in Table 1. It should be clear, throughout the text that MRPSDR=45 and MDR=6 because there is confusion in the way this data is presented throughout the text. The format of the bibliographic citations should be adjusted to the requirements of the journal.
Author Response
We have attached below the comments and also as a word attachment in case it is easier for you to review:
Dear Reviewers,
It is a pleasure for us to have the opportunity to present our results in this journal. The last few years have been hard due to the Covid pandemic but despite everything, we have managed to continue working on invasive pneumococcal disease (IPD). This important disease causes a high number of people affected all over the world and produces serious lifethreatening conditions due in part to the worrying increase in resistance to penicillin and other antibiotics that has occurred in the last few decades. It is crucial, especially in these times in which pneumonia has become very important in our daily lives, to prevent IPD by using vaccines that we have on the market and newer vaccines of broader spectrum are coming. During the SARS-CoV-2 pandemic, COVID patients were treated with antibiotics as a preventive measure for potential co-infections and secondary infections. The effect of this treatment on the antimicrobial resistance rates of S. pneumoniae and even other respiratory pathogens is not fully understood. We consider that our results are interesting since they show the evolution of antibiotic resistance in recent years and allow us to clarify in serotypes, we should focus our preventive efforts on, adapt the vaccination schedule to the epidemiological situation, and focus on vaccine development.
We appreciate and we receive all suggestions with interest to improve our work. We hope that all the answers and changes made are to your liking and we hope that this work will be interesting for publication in this journal.
Best regards
REVIEW:
I consider that the paper can be published in the current format with minimal changes and spelling corrections.
It would be advisable to describe the typology of the samples from which the bacterial clinical isolates were obtained.
We thank the Reviewer for his suggestion. We have added in material and methods the following sentence:
The samples analyzed are mainly blood cultures, although any sterile sample that demonstrates the existence of IPD has been included in the study, such as, for example, cerebrospinal fluid, pleural fluid and joint fluid.
I also recommend including minimal information on the serotype of the 5672 isolates not included in Table 1.
The clinical isolates from this study have been characterized in previous studies published by our group describing the circulating serotypes confirming that in Madrid and Spain, serotype 8 was the most prevalent followed by serotypes 3 and 22F of all IPD cases in adult population although these serotypes are usually fully susceptible (De Miguel S CID 2021; De Miguel S et al, Microorganisms 2021). This sentence has been added in the Discussion section of this manuscript.
It should be clear, throughout the text that MRPSDR=45 and MDR=6 because there is confusion in the way this data is presented throughout the text.
We thank the Reviewer for his suggestion. We have made some changes in the text and this item has been indicated in results:
All MRPSDR (n=45) and/or MDR (n=6) strains belonged to nine serotypes: 19A (n=13), 15A (n=12), 9V (n=12), 14 (n=7), 24F (n=3), 15F (n=1), 19F (n=1), 6B (n=1) and 6C (n=1) (Table 1).
The format of the bibliographic citations should be adjusted to the requirements of the journal
We thank the Reviewer for his suggestion. We have made the changes

Round 2
Reviewer 2 Report
The manuscript can be accepted
Reviewer 3 Report
Thank you for revisiing the article according to my suggestions. I accept it in present form, however the article requires a minor linguistic correction.